# Talking about the Vaccine after the Pandemic: A Cross-Sectional Study among Youth in Turkey and Ethical Issues

**DOI:** 10.3390/vaccines11010104

**Published:** 2023-01-01

**Authors:** Ahmet Özdinç, Mehmet Sait Değer, Muhammed Atak, İbrahim Demir

**Affiliations:** 1Department of History of Medicine and Ethics, Cerrahpasa Medical Faculty, Istanbul University-Cerrahpasa, 34098 Istanbul, Turkey; 2Department of Public Health, Medical Faculty, Hitit University, 19030 Corum, Turkey; 3Department of Public Health, Istanbul Medical Faculty, Istanbul University, 34093 Istanbul, Turkey; 4Turkish Statistical Institute, 06420 Ankara, Turkey

**Keywords:** COVID-19, vaccination, youth, ethics

## Abstract

Examining the factors that affect the vaccination rate among young people in an ethical frame can support vaccination promotion. Therefore, this study will elaborate, through an ethical lens, on young people’s hesitation about and decisions regarding getting vaccinated. The cross-sectional study was conducted with 2428 people aged 15–30 in Turkey in June 2022. The questionnaire included the following subtitles: psycho-social situation, health services and health policies, COVID-19 vaccine, and predictions about life and health after the pandemic. The average age was 22.9 years. In the study sample, 80% were vaccinated, while 20% were not. Vaccinated participants acted more cautiously to protect their health. Receiving accurate and sufficient information on proposed vaccines affects vaccination status. The primary reason for getting vaccinated was “to protect their health, families, and relatives”, and the primary reason for not getting vaccinated was “not trusting the vaccine content or the country where the vaccine was produced”. Specifically, those vaccinated felt more relaxed physically, psychologically, and socially. In addition, the expectations for the future of those vaccinated were significantly higher. Accurate and adequate information is essential for reducing vaccine hesitancy. In addition, promoting prosocial behaviors in young people and highlighting related values will support vaccination.

## 1. Introduction

One of the most essential and cost-effective public health measures is immunization. For example, eradicating smallpox with vaccinations cost $100 million but has saved the world $1.3 billion annually ever since [1]. The effects of immunizations are measured directly by assessing the impact on the vaccinated individual and indirectly by assessing the impact on the unvaccinated community, the epidemiology of the pathogen (modification of serotypes or prevention of epidemic cycles), and the additional benefits resulting from improved health due to receiving the vaccination [2]. For example, in one study, two-thirds of the participants agreed that a vaccine effectively controlled viral spread. At the same time, one-quarter believed they did not need the vaccine if the others were vaccinated [3].

The World Health Organization (WHO) declared the coronavirus-2 (SARS-CoV-2, COVID-19) infection a global pandemic on 11 March 2020 [4]. According to the WHO, as of 3 November 2022, more than 628 million confirmed SARS-CoV-2 infections and more than 6.5 million related deaths were reported in 223 countries [5]. One year after the pandemic was declared, many vaccines appeared on the market [6]. With vaccination campaigns carried out with an intense agenda around the world, within 2 years, 69% of the world’s population had received the first dose of the COVID-19 vaccine, and 63% had received the booster dose. According to a report published on 11 October 2022, this rate reflects 23% (first dose)/19% (booster dose) in low-income countries (LICs) and 81% (first dose)/75% (booster dose) in high-income countries (HICs) [7].

Researchers reported that the COVID-19 vaccine was safe and effective in people with various diseases but noted that many unanswered questions remain on the subject [8]. Many new studies on this subject have increased these questions. For example, some research findings have suggested a link between the COVID-19 vaccination and the development of various cutaneous complications [9,10,11]. However, another study claiming no “zero risk” vaccinations exist uncovered rare cases of thrombosis in patients vaccinated with mRNA COVID-19 vaccines [12]. Despite these examples, emphasis has been placed on the need to be vaccinated and on keeping all patients under surveillance [13].

### 1.1. Vaccination Hesitancy

Health authorities and public resources promote vaccinations. Nevertheless, the public cannot be described as fully accepting of the vaccine. In this context, the reaction of those not fully accepting of the vaccine can be considered as a hesitation rather than a complete rejection. The WHO defines vaccine hesitancy as a “delay in accepting or rejecting vaccines despite the availability of vaccine services” by individuals affected by indifference, conformity (comfort), and/or trust [14]. Edward and colleagues expanded the WHO definition of vaccine hesitancy, noting that individuals who are hesitant about vaccinations may accept all vaccines but remain concerned about them, reject or postpone some vaccines, or refuse all vaccines [15]. Therefore, vaccine refusal is only one dimension of vaccine hesitancy. The concept of vaccine hesitancy is preferred to vaccine refusal also because it has a broader scope, which has led to the recognition, identification, and exploration of a broader spectrum of attitudes toward vaccinations beyond the simplistic and often ideological attitudes that constitute “vaccine rejection” or “vaccine resistance” [16].

Distrust in vaccines is considered a threat to the success of vaccination programs. Vaccine hesitancy can explain a decrease in vaccine coverage and an increase in vaccine-preventable epidemics and risks [17]. Acceptance or rejection of a vaccine is said to result from a decision-making process influenced by a combination of sociocultural, political, religious, and personal factors [18]. Leask classified target vaccine recipients into five groups: unquestioning acceptors, cautious acceptors, hesitant parents, late or selective acceptors, and rejecters of all vaccinations [19]. Streefland et al., on the other hand, defined three attitudes at the point of not accepting the vaccine. The first attitude comes from the person who wants to get vaccinated but cannot. In this case, various structural and material barriers exist. The second attitude is exhibited by those who refuse to get vaccinated. Here, accessibility of vaccination services may be problematic. The third attitude is held by people who question the need for the vaccination [20]. This questioning can develop the individual attitude into a collective attitude. For instance, some groups took a collective attitude of this type against the smallpox vaccination in the 19th century [21]. The main reason for this opposition was the identification of the new technology with the colonial power [22].

Lane and fellow researchers analyzed worldwide vaccine hesitancy based on data from the WHO/United Nations Children’s Fund (UNICEF) joint report published between 2015 and 2017. Three reasons for vaccine hesitancy were cited in the reports most often globally: (1) risk-benefit (scientific evidence), for example, “vaccine safety concerns”, and “fear of side effects”; (2) lack of knowledge and awareness about vaccination and its importance, for example, “families lack knowledge about the benefits of vaccination”; and (3) religion, culture, sex, and socioeconomic issues related to vaccines, for example, “some religious denominations (minority)” and “traditional cultural beliefs.” [23] Religious reasons for refusing immunization generally stem from concerns about vaccine safety or from personal beliefs among a social network organized around a faith community rather than from theologically-based objections per se [24].

In April 2020, a research team from New York University collected data on public health and the psychological and social factors that may be relevant to the COVID-19 pandemic from a sample of more than 50,000 participants worldwide. In light of these data, we examined how conspiracy beliefs about COVID-19 affected public health restraint behaviors and policy support through moral variables. The results revealed that belief in conspiracy theories reduced the adoption of health-related restraint behaviors and policy support for public health measures [25].

In recent decades, despite significant efforts, few public health strategies have been effective against anti-vaccine movements [26]. Thus, existing public health communications about vaccines need to be more effective against these movements [27]. Although providing scientifically based evidence on the risk-benefit ratios of vaccines is important, more is needed to ensure public confidence in vaccines [28]. Vaccine refusal and education have been described as related; those who refuse are more likely to receive a university education than those who accepted the vaccine. Therefore, scientific evidence alone does not affect them and may even increase their determination not to vaccinate [29].

### 1.2. Vaccination Hesitancy and Youth

Dubé et al. grouped anti-vaccine arguments into five categories: distrust of health authorities and healthcare providers, low threat of disease, lack of efficacy of vaccines, lack of safety of vaccines, and alternatives to vaccines [30]. Young individuals are associated with a low threat of disease. For example, COVID-19 follows a course in adults different from the one followed in young people, whereby illness from COVID-19 reportedly is less severe in children than in adults [31].

Moreover, young people mostly survive the COVID-19 disease asymptomatically [32]. At the same time, the contagiousness of social behavior and active life presumably may be high for this age group [33]. From this perspective, and considering the contribution of the vaccine to the prevention of disease transmission [34], the relationship between the young and the vaccine emerges as an essential topic in the fight against pandemic diseases such as COVID-19.

Few cross-sectional studies provide insights into vaccine hesitancy regarding the periods before and after the COVID-19 vaccinations became available. A study was conducted in Switzerland to measure participation in a possible vaccination program in the early stages of the pandemic and to increase incentives afterward [35], while a Swedish study showed that one-third of the participating adolescents aged 15–19 could not decide whether they wanted to be vaccinated [36]. In a study conducted before the vaccine had been released to the market, vaccine hesitancy was found among 345 U.S. high school students aged 12–15 [37]. Similar studies have been published since the introduction of the COVID-19 vaccine. For instance, a study measuring the effect of intrinsic and extrinsic motivations and risk perception among American adolescents uncovered a strong relationship between risk perception and vaccinations [38]. Additionally, the results from research involving healthcare students in Vietnam linked around 40% of the students to vaccine hesitancy, which was attributed to vaccine manufacturers’ failure to disclose the adverse effects of the products and to some students’ belief that possible side effects may result in death [39]. In a cross-sectional study investigating COVID-19 vaccine behavioral intentions among youth in Kenya, the vaccine’s perceived adverse effects were among the critical factors causing vaccine hesitancy [40]. Finally, a study conducted in Nigeria that compared the attitudes toward vaccines of rural and urban young populations revealed that the young rural population was more willing to vaccinate than the young urban population [41].

The cross-sectional studies in the literature suggest that vaccine hesitancy needs to be addressed from the perspective of medical ethics. Considering the issue based on ethical values can affect an individual’s perception of the quality and acceptability of health services. In this study, we examined whether young people in Turkey should be vaccinated and the factors affecting this situation in an ethical framework. This research, conducted among the most dynamic group in society, evaluated situations related to vaccination decisions after the COVID-19 vaccine became available. Only a few studies in the literature have exclusively examined through an ethical lens young people’s hesitation about and decisions regarding getting vaccinated.

### 1.3. Study Hypothesis

Young people are associated with the duration of the epidemic due to their dynamic nature. Therefore, the vaccination rate among young people is considered essential in preventing the epidemic. As such, examining the factors that affect this rate in an ethical frame can support vaccination promotion.

## 2. Materials and Methods

### 2.1. The Population of the Study and Sampling

The target population of the cross-sectional study was individuals aged 15–30 living in Turkey. According to the address-based population registration system data of TURKSTAT, 20,702,872 people in that age range resided in Turkey in 2021 [42], which led to an acceptable sample size calculated at 2401 with a 95% confidence interval and 0.02 margin of error. Hence, the field research was carried out with 2428 young people between the ages of 15 and 30 in Turkey between 11 and 17 June 2022, in 26 provinces determined by the NUTS-2 method, which is the statistical regional unit classification, with the idea that the study findings will be generalizable to all youth in Turkey.

### 2.2. Data Collection Tools

The questionnaire, which was designed based on the literature [43,44,45,46], included the following subtitles: “sociodemographic characteristics”, “psycho-social situation during the pandemic”, “evaluation of health services and health policies during the pandemic”, “outlook on the COVID-19 vaccine”, and “evaluation of predictions about life and health after the pandemic”. While some of the survey questions required “yes” or “no” answers, some were multiple choice, and others required responses using a 5-point Likert scale.

### 2.3. Data Collection Method

This cross-sectional study based on a quantitative research methodology was conducted using the CATI (computer-assisted telephone interviewing) method among the 15–30-year-old population residing in 26 regions in Turkey. The survey took an average of 15 to 20 min to complete. Participants were informed orally about the study, and their consent to participate was obtained before starting the interviews. The data was collected by ADA Research & Consultancy company, certified by the European Society for Opinion and Marketing Research, and ISO 20252.

### 2.4. Statistical Evaluation

The data were evaluated with the SPSS v20.0 (SPSS Inc., Chicago, IL, USA) software package program. The normal distribution of the data obtained by the measurements was evaluated. Descriptive data are presented as a percentile, mean ± standard deviation (SD), or mean. Either the chi-squared or Fisher’s exact test was used for categorical variables according to the suitability of the data for the comparison between groups in the analyses. Analyses for continuous variables were performed with the independent samples t-test or Mann–Whitney U test for two groups and with the ANOVA or Kruskal–Wallis test for more than two groups under normal distribution conditions. Correlation regression analyses were also performed. The significance level was accepted at *p* < 0.05.

## 3. Results

The sociodemographic characteristics of the research participants are presented in Table 1. The average age of the research sample of 2428 people was 22.9 years, and the sample included 1155 (47.57%) female and 1273 (52.43%) male participants. In the group, 1084 people (44.65%) were 19–24 years old, while 23.2% were university students, and 30.6% were high school graduates. Among the participants, 21.3% were married, and 77.2% were single; moreover, 30.8% of the research sample comprised unemployed students, and 37.9% were working but were not students. Analysis of the family demographics indicated that 74% of the participants were part of a nuclear family, 21.1% had an extended family, the family income of 41.4% of the participants was less than USD 350 US, and 22.9% reported a family income of USD 590 or more.

In the study sample, 79.98% were vaccinated. Participants’ reasons for being and not being vaccinated are outlined in Figure 1 and Figure 2. According to these data, 55% of the participants stated that they were vaccinated to protect their health, families, and relatives. On the other hand, 68.7% of the participants who had not been vaccinated stated that they did not get vaccinated or at one point had planned not to get vaccinated because they did not trust the vaccine content or the country where the vaccine was produced, they were afraid of the side effects of the vaccine, and/or they did not have enough information about the vaccine.

### 3.1. Who Has Been Vaccinated?

Table 1 shows the relationship between vaccination status and demographics. According to the statistical analysis, no significant differences existed in vaccination status according to sex (*p* > 0.05), while the highest vaccination rate was found in the 19–24 age group, and the lowest rate was attributed to the 15–18 age group (*p* < 0.01). Increases in the education level and income and in the vaccination rate were related and in the right direction (*p* < 0.001, *p* < 0.001, respectively).

The relationship between being infected with COVID-19 and being vaccinated is shown in Table 2. According to these findings, 83.9% of those who contracted COVID-19, 79.2% of those who did not get infected, and 68.6% of those who were not consciously aware of being sick from COVID were vaccinated (*p* < 0.001).

### 3.2. The Relationship between Vaccination and Anxiety in Youth

The results of the analysis of the relationship between the anxiety experienced during the pandemic (before the COVID-19 vaccinations became available) and the vaccine are provided in Table 2. The existence of a relationship between the leading cause of anxiety experienced during the pandemic process and an individual’s vaccination status was investigated, revealing that the anxiety levels of those who were vaccinated were approximately 10% higher than the anxiety levels of those who were not (*p* < 0.001). Vaccinated young people experienced more anxiety than those not vaccinated, such as being anxious about catching COVID-19 or about dying from the disease, thinking that family members and relatives could catch or die from COVID-19, and the uncertainty of the pandemic process (*p* < 0.001).

The ability to cope with thoughts about COVID-19 during the pandemic (Table 2) was examined with the chi-square independence test according to whether the individual was vaccinated, which revealed a significant relationship between being vaccinated and coping with thoughts about COVID-19 (*p* < 0.001). Those vaccinated seemingly could not easily resist thoughts about the disease (44%) and had less difficulty resisting these thoughts (33.7%). In contrast, those who were not vaccinated could easily resist thoughts about the disease (55.3%) and had less difficulty resisting these thoughts (22.6%).

Whether the young people felt more comfortable physically, psychologically, and socially compared to the way they felt during the height of the pandemic period was examined with the independent sample t-test according to the participant’s vaccination status, and a significant difference was found, as shown in Table 3 (*p* < 0.001). Specifically, those who had been vaccinated felt more relaxed physically, psychologically, and socially than those who had not been vaccinated.

The independent sample chi-square test was used to determine whether the effect of the pandemic process on young people’s expectations for their futures differed according to their vaccination status; a significant difference was found between vaccinated and unvaccinated participants (*p* < 0.05), demonstrating that the expectations for the future of those who were vaccinated (11.5%) were significantly higher than the expectations of those who were not vaccinated (8.2%) (Table 2).

### 3.3. Access to Vaccine and Overview of Applied Vaccine Policy

As illustrated in Table 2, the chi-square independence test was employed to determine whether a relationship existed between being vaccinated and not have difficulty accessing healthcare services. The results showed that those who were vaccinated had more comfortable access to healthcare services than those who were not vaccinated (*p* < 0.001). While the rate of those who stated that they could access health services among those vaccinated was 59.5%, the rate of those who not vaccinated was 42.2%.

It has been determined that receiving accurate and sufficient information on proposed vaccines during the pandemic affects vaccination status (*p* < 0.001). The rate of being vaccinated (84.5%) for those who thought they had enough information was significantly higher (as shown in Table 2) than the rate for those who believed the information they had received was insufficient (15.5%).

In addition, the relationship between information from the MoH and the scientific committee and vaccination status was examined, the results of which are depicted in Table 2. The rate of vaccination (83.9%) of those who found the information from the MoH and the scientific committee about the disease and the recommendations to combat the pandemic to be sufficient and correct was significantly higher, by 9 points (*p* < 0.001), than the rate of those who found the MoH and scientific committee information to be lacking.

The study results identified reasons for being vaccinated, which were associated with receiving accurate and sufficient information about the pandemic. As shown in Table 4, 71.8% of those who believed in the accuracy of the positive results related to vaccinations announced by the MoH thought that they had received correct and sufficient information about the pandemic. On the other hand, 33.6% of those forced to get vaccinated by their friends and relatives did not think they had received accurate and sufficient pandemic-related information.

The data also revealed a relationship between satisfaction with health services and vaccination. This situation, which was examined with the Pearson correlation coefficient, revealed a low negative relationship between satisfaction and vaccination (*p* < 0.001). This means that those who were not vaccinated were less satisfied with health services than those who were vaccinated. A relationship also existed between satisfaction with health services and seeing vaccine policy and advocacy as correct and adequate. These data were examined with the Pearson correlation coefficient, and a moderately positive relationship was found between them (*p* < 0.001): those who thought the vaccination policy was correct and its advocacy was sufficient were more satisfied with health services.

### 3.4. Attitude toward Health Protection

Whether the behavior of young people to protect their health differs according to their vaccination status was examined with the independent sample t-test, and a statistically significant difference was found (*p* < 0.05). Table 3 presents the results that indicate those who were vaccinated acted more cautiously to protect their health than those who were not vaccinated.

The research also uncovered that the frequency of drug use for the young people included in the study increased more in those who were vaccinated (83%) than in those who were not vaccinated (13%) after the pandemic (*p* < 0.001). In the analysis conducted to determine the focus that guides this use, individuals who had been vaccinated were found to follow more information about the use of supplements gleaned from social media platforms than from a pharmacist or doctor during and after the pandemic. The proportion of these individuals (76%) was about three times greater than the proportion of those who were not vaccinated. This ratio was statistically significant, and the analysis was tested with binomial statistics (*p* < 0.001).

## 4. Discussion

Four out of five young people participating in the study indicated they had been vaccinated. According to a report dated 11 October 2022 [7], this rate is about the same as the rate for HICs and above the world average based on the whole population. Nearly half the people who hesitated to get vaccinated, in the range of 20–40% reported in the literature [39,40,47,48], accepted the vaccine later. As a dimension of vaccine hesitancy, the presence of individuals who both accept the vaccine and maintain their concerns or postpone vaccination due to their concerns [15] may be in question in this study. Notably, this research was conducted at the final stage of the COVID-19 impact and vaccination initiative. Therefore, comparing current vaccination rates with those from the previous vaccine hesitancy studies suggests the “postponement” emphasized in the definition of vaccine hesitation was occurring [14,15]. In general terms, not every hesitation results in total refusal to get vaccinated.

In the literature, women have been shown to be more hesitant than men about vaccinations [49]. However, our study found no significant difference in vaccination status by sex. Therefore, when these two results are compared, the conclusion can be drawn that females are more hesitant about vaccinations than males but that does not prevent them from demonstrating the same attitude as males at the end of the process.

When the relationships between income status and education level and vaccination status were examined, the claim in the literature that poverty and lack of education strengthen vaccine hesitancy [49]. We obtained a similar relationship between low income/education status and not being vaccinated in our study. Furthermore, the vaccination rate was directly proportional to education level and income group. Accordingly, income and educational status have a positive effect on vaccination.

Some authors attributed the limited coverage of the vaccine to the supply difficulties and the negative news produced on social media [50]. Our study determined that accessibility also was related to not being vaccinated. Although this situation is reported in the literature, especially in areas where accessing the vaccine is difficult, such as African countries [50], the findings showed that those who stated they had easy access to health services were more often vaccinated. Accessibility to health services, therefore, was directly related to vaccination rates.

The reasons for hesitating to and deciding not to get vaccinated uncovered in our study parallel those mentioned in the literature [51,52]. Lack of confidence and knowledge is the main reason for vaccine hesitancy and non-vaccination. Numerous studies on vaccine safety that prompt hesitation about side effects [9,10,11,12] encourage the answers “I am afraid of the side effects” and “I do not trust the content of the vaccine”, which are among the reasons for not being vaccinated we found. The anti-vaccine “low disease threat” and “ineffectiveness of vaccines” arguments claimed by Dubé et al. [30] are similar to the answers “not being thought to be protective” and “belief that the immune system will overcome the disease” in our study. In our research, “not trusting the country where the vaccine was produced”, as one of the notable dimensions of distrust, coincides with the claim that “the vaccine was introduced by colonial powers” in the first anti-vaccine movements [22].

The concerns about the disease are dominant in the direction of being vaccinated. In our study, being afflicted with the COVID-19 disease stands out as support for vaccination. The repetition of anxiety caused by a bad experience with a previous disease seems to support the attitude toward being vaccinated.

Our research revealed a significant relationship that sought to answer whether the vaccine effectively relieved concerns about the pandemic and COVID-19 disease. Accordingly, those vaccinated were more physically, psychologically, and socially comfortable and had higher expectations for their futures than their unvaccinated counterparts. Thus, we can say that being vaccinated may have a strong relationship with reducing anxiety during the pandemic. This implication is in line with other research [53,54].

### 4.1. Can Vaccination Rates Be Increased?

The vaccination policies of states and governments are among the critical factors affecting vaccine hesitancy and anxiety [55]. The public health system can influence vaccine acceptance primarily through three initiatives: implementing an inclusive vaccination policy, developing and implementing immunization recommendations, and vaccine safety monitoring [56].

To increase vaccine acceptance, first, structural barriers, such as the lack of transportation, finances, or assistance for each individual, must be minimized [29]. This recommendation is supported by the relationship between access to health services and getting vaccinated in our study, as our data evidence that vaccinated people in the sample had more access to health services than those who were unvaccinated.

The second way to increase vaccine acceptance is to devise effective and appropriate communication strategies. Through our study, we discovered that receiving accurate and sufficient information during the pandemic strongly affected vaccination. Furthermore, in communications about vaccination, consistency, transparency, and community-based negotiation can increase trust [57]. In this context, attention is drawn to two main topics: disseminating correct information and combating false information.

The positive effect of improving health literacy among young people related to vaccination attitudes was emphasized in the results [58]. In this context, some innovative methods have been proposed that should go beyond open communication models suitable for people’s experiences and social environments. For example, a critical study has demonstrated that applications such as chatbots reduce vaccine hesitancy [59]. On the other hand, a significant change occurs in the intention to receive the vaccine after receiving information via video compared to receiving it in text [33]. Some authors also argue that next-generation messages designed to increase vaccination should consider religious beliefs more carefully [60].

Access to various sources that hold information about vaccines, especially those providing inaccurate information, affects decision-making [56]. Some scholars have concluded that people’s perceptions of vaccine acceptance were influenced by the flow of information on various social media platforms and by the severity of COVID-19 cases [3]. A study in this context found that the more the participants were exposed to statements about the severity of post-COVID-19 vaccine side effects on social media, the more severe their own post-vaccination side effects were [61]. Emphasizing the spread of significant amounts of misinformation and news about COVID-19 on social media, researchers have highlighted the use of fear-mongering and fake experts and the implementation of strategies supported by psychology that increase the ability to identify more honest or reliable news against conspiracy theories [62,63].

Our research revealed that the statements made by public authorities (Ministry of Health (MoH), Scientific Committee) contributed to increased vaccination. Although this method inspires confidence, the amount and frequency of information about the COVID-19 illness and vaccines that these officials shared is a matter of debate. Some studies have found that overprotective behavior and frequent access to news about COVID-19 added to public concern [64].

Healthcare professionals must actively contribute to vaccination campaigns: this is the last method recommended to increase vaccine acceptance [65,66]. The critical role that healthcare professionals who treated patients at the potential cost of their own lives during the pandemic play in managing fear and widespread vaccine rejection was also emphasized [29]. Although this recommendation seems reasonable, vaccination hesitancy among healthcare professionals must be considered to determine exactly how reasonable such a recommendation is. A study conducted to detect vaccine hesitancy among healthcare professionals in the United States revealed that almost half of the nurses had hesitations about the COVID-19 vaccine; this rate was around 13% among physicians [67]. This result supports the need for physicians to increase their efforts to keep people safe and informed about the COVID-19 illness and vaccines.

### 4.2. Dealing with Ethical Issues

The ethical debate about vaccination is based on the morality of the incentive or coercion to be vaccinated or not to be vaccinated. On these grounds, Beauchamp and Childress put forward an evaluation based on the principles of being beneficial, not doing harm, respecting autonomy and justice [68]. Information about the benefits of vaccines has been presented throughout history [69]. Today, vaccines are associated with eradicating some epidemic diseases that no longer exist in the world [70,71]. The world health authorities give positive messages regarding the COVID-19 vaccines that entered the market at the end of 2020. However, both doubts about its benefits and some evidence of a violation of the principle of doing no harm [72] have created an ethical dilemma.

Some discussions about the vaccine issue can be handled within the principle of respect for autonomy [73]. The term “autonomy” means having a voice and power over oneself, without being under the control of others and without being in a situation that prevents one from making meaningful choices [68]. The principle of autonomy must meet four mandatory criteria for its legitimacy: the patient’s decision-making capacity, adequate information sharing, an adequate understanding of information, and voluntariness [74].

The scope and limits of the principle of respect for autonomy in the fight against global pandemics are controversial. Many countries across the world, with a pragmatic point of view, emphasized the benefits of the vaccine rather than its harms and pressured their citizens into compulsory vaccination [75]. Some authors suggested complicating the procedure for refusing the vaccine to support this approach [76]. In addition, the violation of autonomy can be justified by arguing that the public interest (public health) “is the intimate and ultimate determining factor of individual freedom and is above any individual benefit” [77]. However, these approaches cannot be easily justified through an ethical lens.

Among the issues to be considered regarding ensuring transparency, which is one of the components of autonomy, is granting full access to vaccination records to parents and the general public [78]. Again, ethical problems emerge related to the vaccination of individuals who cannot make decisions in the context of autonomy. The public health and public interest argument is more open to debate for those who lack capacity because of its scope and potential for exploitation [79].

The issue of fairness in the distribution of resources is another prominent area of discussion during a public health emergency. The issue of allocation is one of the crucial challenges of medical ethics. The report on vaccine use showed that the vaccination rate of HIC countries was four times the vaccination rate of LIC countries [7]. On the other hand, some countries have made mandated that students be vaccinated against COVID-19 in order to attend school [80]. A potential ethical problem arises concerning unequal access to resources in both approaches.

Of course, conducting ethical discussions pertaining to a global pandemic that suddenly affects all people on Earth in a short time is not easy. Indeed, the argument has been made that a consistent attitude that requires precise decision-making based on scientific, economic, or moral considerations cannot be agreed upon in the context of limited time [81].

### 4.3. Strengthening Moral Behavior

It is claimed that there is a strong relationship between moral concerns and positive behaviors [82]. Besides that, recognizing the moral components of a subject constitutes the basis of moral behavior [83,84]. Regarding communicable diseases, moral motives contribute to increased vaccinations when there is awareness of the moral aspects of vaccinations [85,86]. Some studies support this claim. For example, a study among university students in Australia highlighted the strong relationship between inoculation and the perception of moral duty [87].

In our study, the goal of protecting their own health and the health of friends and relatives was among the reasons for vaccinating young people who had been vaccinated. From this, we can propose a stronger emphasis on altruism as a value for vaccination promotion. Altruism prompts individuals to develop deliberate and voluntary action where the primary goal is to improve the well-being of another person [32,88]. In this context, encouraging prosocial behaviors [88], in which people are highly motivated to help strangers they may never meet again, donate their time and goods to charities, and take care of their friends and family, is an excellent method for overcoming vaccination hesitations.

### 4.4. Limitations of the Study

In this research, the age range determined for the group referred to as “young” people was 15–30. This range may be expanded or narrowed in other sources [89,90]. Since this is a cross-sectional study, precise causal inferences cannot be made. The available conclusions do not necessarily form about the direction of the relationships between the variables. Future longitudinal and/or experimental studies may shed more light on the relationships between the variables studied here.

## 5. Conclusions

Immunization is one of the essential tools for preventing pandemic diseases. Reducing vaccine hesitancy during a global pandemic and realizing rapid and simultaneous vaccination will reveal the success of preventing the spread of a pandemic. Young people, the most dynamic group in society, can be good partners for public health professionals working on pandemic prevention. Reducing young people’s anxiety during the pandemic with accurate and adequate information will strengthen this partnership. In addition, promoting prosocial behaviors in young people and highlighting related values will indirectly support vaccination.

## Figures and Tables

**Figure 1 vaccines-11-00104-f001:**
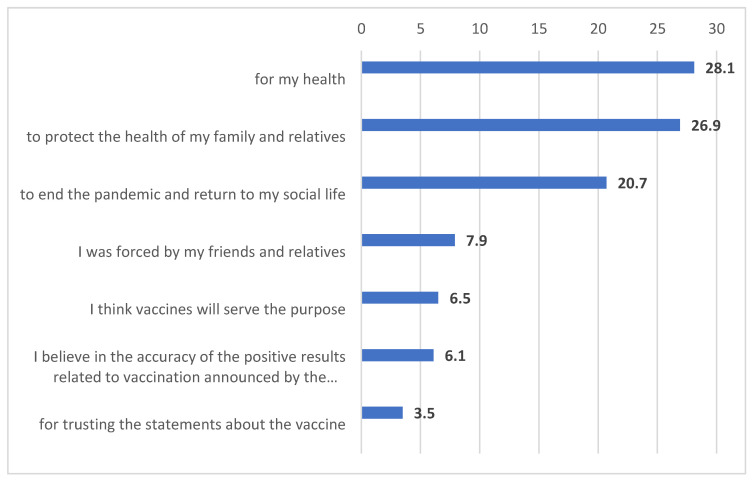
Participants’ reasons for being vaccinated.

**Figure 2 vaccines-11-00104-f002:**
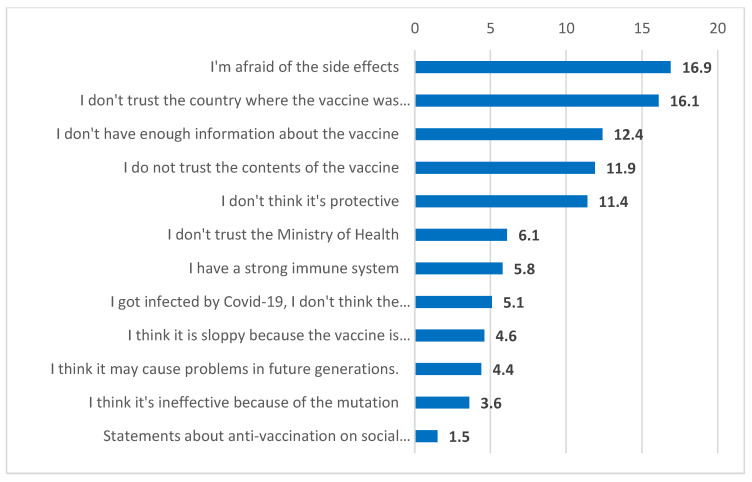
Participants’ reasons for not being vaccinated.

**Table 1 vaccines-11-00104-t001:** The sociodemographic characteristics of the research participants and vaccination status.

	Total		Vaccination Status			
			Yes		No	
	(n)	(%)	(n)	(%)	(n)	(%)
Total	2428	100	1942	80	486	20
Sex						
Female	1155	47.57	928	47.8	227	46.7
Male	1273	52.43	1014	52.2	259	53.3
Age						
15–18	438	17.96	295	15.2	141	29.0
19–24	1084	44.65	945	48.7	139	28.6
25–30	908	37.40	702	36.1	206	42.4
Place of birth						
Village-Town	136	5.60	106	5.5	30	6.2
District	795	32.70	649	33.4	146	30.0
Province	1019	42	787	40.5	232	47.7
Metropolitan area	465	19.20	389	20.0	76	15.6
Abroad	13	0.50	11	0.6	2	0.4
Marital status						
Married	516	21.3	394	20.3	122	25.1
Single	1875	77.2	1518	78.2	357	73.5
Divorced	37	1.5	30	1.5	7	1.4
Educational status						
Primary school graduate and below	79	3.3	52	2.7	27	5.6
Secondary school student	76	3.1	48	2.5	28	5.8
High school student	308	12.7	195	10.0	113	23.3
University student	563	23.2	516	26.6	47	9.7
Graduate student	39	1.6	33	1.7	6	1.2
PhD student	1	0.00	0	0.0	1	0.2
Secondary school graduate	170	7	125	6.4	45	9.3
High school graduate	742	30.6	578	29.8	164	33.7
Graduated from a university	407	16.8	360	18.5	47	9.7
MSc	40	1.6	33	1.7	7	1.4
PhD	3	0.1	2	0.1	1	0.2
Working status						
Student	749	30.8	604	31.1	145	29.8
Student and working	288	11.9	232	11.9	56	11.5
Student not working	921	37.9	739	38.1	182	37.4
Neither student nor working	470	19.4	367	18.9	103	21.2
Family type						
Nuclear family	1797	74	1465	75.4	332	68.3
Extended family	512	21.1	379	19.5	133	27.4
Broken family	119	4.9	98	5.0	21	4.3
Household income (monthly)						
USD 349 and below	1005	41.4	763	39.3	242	49.8
USD 350–589	720	29.7	569	29.3	151	31.1
USD 590 or more	555	22.9	485	25.0	70	14.4
No answer	148	6.1	125	6.4	23	4.7

**Table 2 vaccines-11-00104-t002:** The relationship between being infected with COVID-19, anxiety experienced during the pandemic, ability to cope with thoughts about COVID-19, effect of the pandemics on young people’s expectations for their futures, accessing healthcare services, receiving accurate and sufficient information on proposed vaccines, information from the MoH and the scientific committee and being vaccinated.

		Vaccination Status	
		Yes	No
infected with COVID-19	Infected	83.9%	16.1%
	Non-infected	79.2%	20.8%
	Don’t know (didn’t get tested)	68.6%	31.4%
anxiety experienced during the pandemic	My anxiety level hasn’t changed compared to the pre-pandemic period	33.2%	42.4%
	My anxiety level has changed compared to the pre-pandemic period	18.7%	20.0%
	My anxiety level has increased slightly compared to the pre-pandemic period	42.9%	32.5%
	My anxiety level has increased to the level of using antidepressant medication	5.1%	5.1%
ability to cope with thoughts about COVID-19	I could easily resist thoughts about the disease	44.0%	55.3%
	I had a less difficulty to resist thoughts about the disease	33.7%	22.6%
	It was very difficult for me to resist the thoughts of the disease	18.4%	18.7%
	I received psychological support to resist thoughts about the disease	3.9%	3.3%
effect of the pandemics on young people’s expectations for their futures	My expectations for my future have decreased	44.7%	43.0%
	My expectations for my future have not changed	43.8%	48.8%
	My expectations for my future have increased	11.5%	8.2%
accessing healthcare services	I have never been able to access health services	9.1%	9.5%
	I couldn’t access health services	9.4%	15%
	Partially accessed health services	22%	33.3%
	I was able to access health services	24%	20.2%
	I was able to access health services very easily	35.5%	22%
receiving accurate and sufficient information on proposed vaccines	I definitely don’t think I received accurate and sufficient information	81.2%	18.8%
	I don’t think I received accurate and sufficient information	74.5%	25.5%
	Uncertain	74.5%	25.5%
	I think I received accurate and sufficient information	81.2%	18.8%
	I definitely think I received accurate and sufficient information	84.5%	15.5%
information from the MoH and the scientific committee	Sufficient and correct	83.9%	16.1%
	Non-sufficient and not correct	75.0%	25.0%
	Not sure	77.4%	22.6%

**Table 3 vaccines-11-00104-t003:** The relationship between feeling physically, psychologically, socially relaxed and behavior to protect health and vaccination status.

	Vaccination Status	n	Mean	SD	df	t	p
Feeling physically, psychologically, and socially relaxed	Yes	1942	3.51	1.308	2426	3.679	0.000 ^*^
	No	486	3.27	1.331			
behavior to protect health	Yes	1942	3.53	1.25	2426	4.635	0.000 ^*^
	No	486	3.23	1.28			

* *p* < 0.05 independent sample *t*-test.

**Table 4 vaccines-11-00104-t004:** The relationship between reasons for being vaccinated and receiving accurate and sufficient information about the pandemic.

	I definitely don’t think I received accurate and sufficient information	I don’t think I received accurate and sufficient information	Uncertain	I think I received accurate and sufficient information	I definitely think I received accurate and sufficient information
To end the pandemic and return to my social life	8.7%	11.6%	21.2%	30.7%	27.8%
To protect the health of my family and relatives	8.7%	9.8%	18.4%	31.1%	31.9%
For my health	8.7%	10.0%	18.1%	27.4%	35.9%
I believe in the accuracy of the positive results related to vaccination announced by the MoH	6.4%	10.9%	10.9%	29.7%	42.1%
For trusting the statements about the vaccine	9.5%	16.7%	11.9%	34.9%	27.0%
I think vaccines will serve the purpose	6.0%	14.4%	20.4%	29.6%	29.6%
I was forced by my friends and relatives	19.1%	14.5%	17.9%	25.2%	23.3%

## Data Availability

The survey dataset is available on https://www.hayatvakfi.org.tr, accessed on 28 December 2022.

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
