# Peer review of "Talking about the Vaccine after the Pandemic: A Cross-Sectional Study among Youth in Turkey and Ethical Issues"

_vaccines, 2023, doi:10.3390/vaccines11010104_

Round 1

Reviewer 1 Report

Dear Editor,

In this manuscript the authors describe an initiative focused on knowing which is the attitude of young people towards vaccination and the COVID-19 pandemic, including reasons for hesitation, such as ethical concern. The article is well written, although certain aspects would need to be solved before acceptance. Congratulations.

MAJOR COMMENTS

-          The Introduction, although interesting, it is way too large. I would suggest to condense it in 3-4 paragraphs.

-          In my opinion, points 2.1, 2.2, and 2.3. are not really methods, but Introduction. You are reasoning why do you perform your survey, but not really how, which is what it has to be covered in Methods

-          Some sentences in Methods come from external sources, please include the respective reference

-          “The questionnaire, which was designed based on the literature” Which literature?

-          How were participants selected?

-          Limitations / Point 2.8.: This should be the pre-last paragraph of Discussion, with a bit longer extension

-          Please, check not to leave any quotation to other publications in the Discussion without a reference

MINOR COMMENTS

-          I would suggest to merge some of the tables. Additionally, in certain cases, it does not make any sense to provide such info as a table and 1 or 2 sentences can make it too

-          In case you want to provide the dataset, which is extraordinarily great, I would suggest to include the meaning of the codes. In its current version I doubt it can be of any use to anyone

-          You are describing (biological) sex (female, male), not (sociological) gender (woman, man, other). Please, use “sex”, not “gender”

-          No need to do a duplication of the tables in the text, just type the most relevant/important/interesting parts

-          I would suggest to include a mention to the equivalence of Turkish liras to a larger currency, or to use a larger currency

-          Is it possible to merge some of the categories in Education? Like “Maximum education achieved”.

-          Just p value is enough, no need for X² value or df. Additionally, it would be great if you could include it in all the rows as appropriate

-          Discussion can be condensed

Reviewer 2 Report

This is an interesting and well-written paper. There is, however, one concern I have regarding data collection. Researchers conducted field research in Turkey between June 11 and 17, 2022, with 2,428 young people between the ages of 15 and 30, in 26 provinces categorized by 172 NUTS-2 statistical regional units. They also mentioned in the methodology that this cross-sectional study was based on a quantitative research method using CATI (computer-assisted telephone interviewing) among the 15–30-year-old population in 26 Turkish regions. It took an average of 15 to 20 minutes to complete the survey. Is it possible for 2,428 young people (between June 11 and 17, 2022) to be interviewed through the CATI method in just seven days? This needs to be clarified since the survey took an average of 15 to 20 minutes to complete, and participants were informed orally about the study before they consented to participate.

Author Response

Dear Reviewer,

Thank you for the review and comment. This data was collected by ADA Research & Consultancy company which is certificated by ISO 20252, the European Society for Opinion, and the Association of Turkey Marketing Research and Researchers. The payment made to the company was funded by Erasmus+ Youth Participation Activities (KA154-YOU) program, Project number KA154-YOU-24A46AFB. The website of the company is https://www.adaarastirma.com/en/

I added this information now to the “Data collection method” part.

Thank you for warning me

Sincerely yours.

Round 2

Reviewer 1 Report

Authors have incorporated all the suggestions made by reviewers. I suggest manuscript acceptance. Still, suggesting a merging of tables as follows:

TABLES

> Table 1 looks fine

> New Table 2 = Old Table 2 + Old Table 3 + Old Table 4 + Old Table 5 (extra column just with p value) + Old Table 6 + Old Table 7 + Old Table 8 + Old Table 9 + Old Table 12 (as with Old Table 5)

> New Table 3 = Old Table 10

> New Table 4 = Old Table 11 (This could be even removed from Table and just leave it as text)

SUPPLEMENTARY TABLES

In its current view, it is not really useful: name of variables is missing, meaning of answers too...

Author Response

Dear Reviewer

Thank you for your decision and suggestion

We have merged table-2,3,4,6,7,8,9 as TABLE-2, table-5,12 as TABLE-3, and table-10 as TABLE-4. In this way, we have just 4 tables
It's much better. Thank you for the suggestion.

The revised dataset file (with names of variables) has been uploaded.

Best regards